# Re-assessing the role of culture on the visual orientation perception of the rod and frame test

**Chéla R. Willey**[¤], **Zili Liu***

Department of Psychology, University of California, Los Angeles, Los Angeles, California, United States of America

¤ Current address: Department of Psychology, Loyola Marymount University, Los Angeles, California, United States of America

* zili@psych.ucla.edu

## Abstract

In recent research of culture and ethnicity in visual perception, some researchers have found support for the hypothesis that more collectivistic cultures tend to be more influenced by surrounding contextual visual information than more individualistic cultures. This hypothesis suggests that even low-level visual perception may not be universal and has great implications on how vision research should be conducted. The current study reexamines this hypothesis in the rod and frame task, which tests the influence of a tilted contextual frame on orientation perception of the rod. We found no difference between participants of East Asian and Western European descent in this task. Despite not finding the cultural effect, we found a well-reported gender effect in which women were more influenced by the tilt of the frame than men, helping to ensure the quality of the data collected. Our results suggest that contextual influence on visual perception does not affect East Asians and white Western Europeans differently.

## 1. Introduction

Can culture influence cognition and perception? The answer appears to be mostly affirmative in the psychological research literature (see [1] for a recent review). A main focus in cultural psychology research has been comparing between people of North America / Western European and of East Asian descent, which putatively represent prototypical individualistic and collectivistic cultures, respectively. Some have posited that these two broad cultures differ even in basic visual perception [2]. Specifically, the perception of an object is assumed more influenced by its spatial context for Easterners than for Westerners. For example, when viewing photographs of natural scenes, American participants were found to fixate on the focal object more quickly and more often than their Chinese counterparts, who made more saccades to the background [3]. The theory behind such findings is the claim that Westerners tend to perceive analytically and independently of context, whereas Easterners perceive holistically and take context into consideration [4]. Such a claim basically predicts that

**Funding:** The author(s) received no specific funding for this work.

**Competing interests:** The authors have declared that no competing interests exist.

Westerners are less vulnerable than their Eastern counterparts to visual illusions, since a large number of visual illusions are context-dependent. These cultural effects may be also lessened as participants of one culture have immersed themselves in the other's culture for sufficient time [5]. One such classic illusion is called the Rod and Frame Test (RFT), and [6] found that East Asians were more error prone than their Western counterparts in judging whether a rotated rod inside a tilted square was vertical or not. They suggested that the perception of the rod's orientation was influenced more by the tilted frame for Easterners than for Western European Americans.

To appreciate the impact of this theory on culture and brain so far, recall that a fundamental assumption in visual neuroscience research is that basic visual functions such as orientation perception are the same for humans and non-human primates alike. If, however, even basic visual functions depend on the perceiver's cultural background, then all studies across the world, in the literature and in the future, need to be considered in the context of the cultures of the participants. Perhaps more importantly, vision research with animals will have much limited utility, which may in turn influence public policies. The aim of the current research is to re-investigate cultural differences of those living in the United States (either in their lifetime or recent international arrivals) on the RFT, a task that measures the influence of a contextual frame on basic orientation perception.

We chose the RFT in part because of the following review article. [7] reviewed evidence of differences between East Asian and Western cultures affecting brain structure and functions. They concluded that there was limited evidence of any structural differences anywhere in the brain, but there was relatively more evidence in neural functional differences in the visual ventral cortex that was associated with perceptual processing. By using a visual perceptual task in the current study, therefore, we targeted a function of the brain that would most likely yield an effect in culture, particularly given that the prior study by [6] had already found such an effect. In addition, using a task such as the RFT made it possible for the context, a tilted frame here, to be objectively and precisely defined.

In an RFT task without cultural considerations, the subjective vertical (SV) of visual orientation, as estimated by the rod orientation, has been shown to be influenced by the spatial context. The original RFT was developed by [8], who presented a luminous rod at the center of a luminous frame, both of which could be independently positioned and rotated. The participant's task was to verbally instruct an experimenter to continue to rotate the rod until they felt that it was vertical, while the frame remained stationary. The results showed how the influence of a tilted frame biased the SV estimate towards the tilt of the frame. Recently, other researchers have adapted the original RFT into different settings, such as 2D computerized tests [9] or 3D virtual reality (VR) environments, the latter of which yielded similar results to the original tests [10]. We have adapted the RFT to display in a virtual reality environment for use in a head-mounted display in 3D.

To preview, in the current study we tested and found the typical bias caused by a tilt of the frame in the RFT. We further found a statistically significant difference between male and female participants, an effect that had been found repeatedly in the literature (e.g., [11–14]). These effects indicated reasonable quality of the data. The average biases towards the tilt of the frame in this study were within the range of typical biases found using a physical rod and frame apparatus. However, our results did not support the hypothesis that East Asians were more error prone than white Americans. In fact, our East Asian participants erred slightly less, yet with statistical significance, than their white counterparts. Our overall finding was that visual perceptual context as used in the RFT did not influence differentially between East Asian and Western participants.

## 2. Method

### 2.1. Participants

All procedures were approved by UCLA's Institutional Review Board. The study was determined to be of minimal risk and the need for written consent was waived. Participants' consent was obtained verbally after they had read through an information sheet that outlined the procedures, benefits, and risks of the study in accordance with the Belmont Report and the Declaration of Helsinki. Consent was documented on a password protected spreadsheet with personal identifiers removed. From the diverse cultural backgrounds of the psychology undergraduate students at the University of California Los Angeles (UCLA), 342 participants completed the RFT alignment task and a cultural questionnaire. Of these participants, 216 also completed the RFT discrimination task. Only four participants completed the RFT discrimination task without the alignment task.

All stimuli were presented to the participants using the Oculus DK1 headset. All stimuli were made using the Unity software. Participants used the mouse buttons to indicate all responses. For all tasks, participants stood with their feet together on a 20 cm high-density foam mattress while performing the RFT. The use of the mattress was meant to increase standing instability, which has been shown to increase SV errors and variability in healthy participants [15].

### 2.2 The alignment task

In this task of adjustment, participants rotated the virtual 3D rod to align with their perceived SV using mouse buttons. They completed six alignment estimates to vertical for each of three tilted frames conditions: -18˚, 0˚, +18˚. Averages and standard deviations were recorded for each participant for each frame tilt. The tilt of the frame was randomly ordered. The initial rod orientation was randomly selected at the beginning of each trial between ± 25˚. Upon seeing the rod and frame, participants used mouse buttons to rotate the rod in 0.2˚ increments until they believed that it was vertical. Once completed, the participant verbally confirmed that the rod was vertical, and the experimenter pressed the enter key to move on to the next trial. A blank gray screen was shown between trials for 1.5 seconds before the next stimulus was presented. In the literature, this method had been used often due to its ease of implementation and robust results. Thus, we were able to compare our results with those in the literature. Additionally, we used it as a preliminary assessment to set up the parameters in the discrimination task, as follows.

### 2.3 The discrimination task

This task used the method of constant stimuli, to obtain a full psychometric function so that both bias and discrimination sensitivity could be obtained. We first used the alignment task to help preset rod orientations for the discrimination task. The average SV estimates from six alignment trials for each tilt angle of the frame were used to determine the preset rod angles for the discrimination task. For each of the tilted frame angles (-18˚, 0˚, 18˚), rod orientations were chosen to be ±1˚, 2˚, 3˚, and 4˚ away from their average SV estimate made during the alignment task (see Fig 1). For the 0˚ frame angle, because of the expected higher discrimination sensitivity, we chose rod orientations that were ±0.5˚, 1˚, 2˚, and 3˚ away from their average SV estimate made in the 0˚ tilt alignment task. For each of the three frame tilt angles, each of the eight rod orientations was presented eight times, resulting in 192 total trials. The presentation order of angles and frame orientations were all randomized. In each trial, the rod and frame stimulus was presented for 250 ms before the screen returned to the

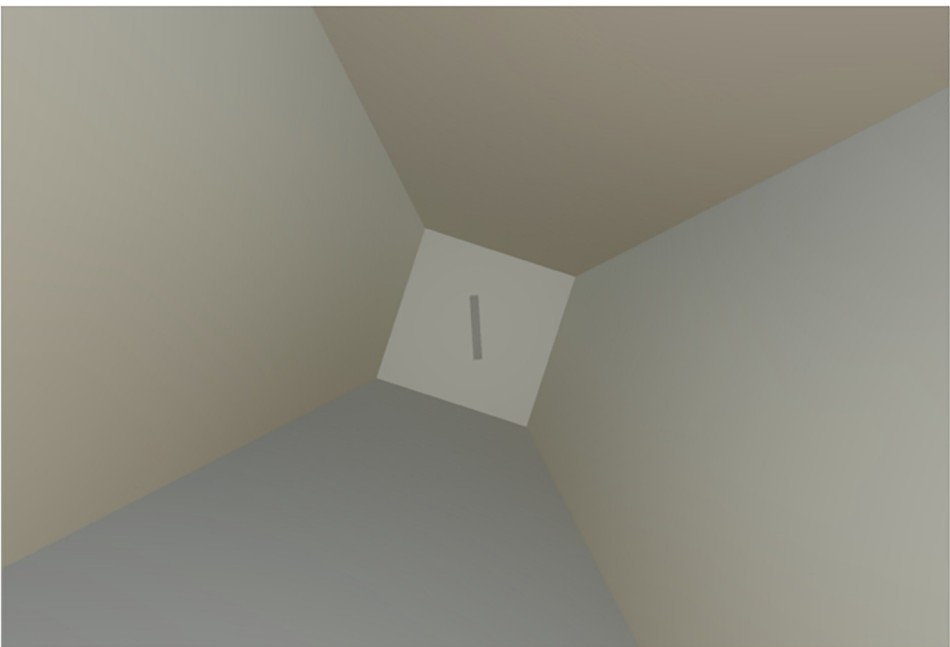

**Fig 1. Participant's view of an RFT trial.** The frame is tilted 18˚ and the rod 0˚ in this example.

background gray. Participants had 2 seconds to respond with a right or left mouse click to indicate whether the rod was tilted right or left from vertical. Even if the participant responded before the 2 second period was over, the screen continued to be blank for the remaining time (see Fig 2).

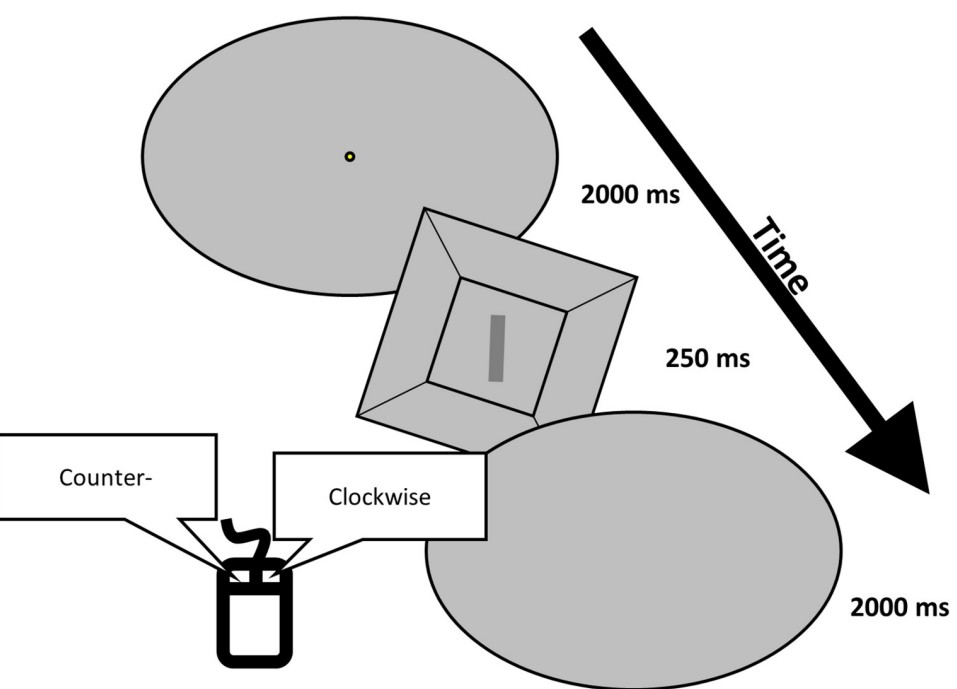

**Fig 2. Schematic of a single orientation discrimination trial in the discrimination task.**

We fitted each participant's data with a cumulative Gaussian using the *psignifit* function [16] in MATLAB (MathWorks Inc., Natick, MA). From this fitting, the angle that was responded to 50% clockwise was the point of subjective equality (PSE), which is defined as the bias here (away from 0˚). The slope of this function at the PSE is the discrimination sensitivity and is the inverse of the standard deviation of the Gaussian. Each participant's data was individually inspected, using the goodness of fit $R^2$, in comparison to the fitted Gaussian to determine if the function adequately fit the data. Participants who responded seemingly randomly or uniformly throughout the experiment were not used in data analyses ($n$ = 15) since the cumulative gaussian function would be unable to capture the psychometric function depicting their perception of subjective vertical. More specifically, if the participant responded counterclockwise or clockwise uniformly on all trials, or if the participant responded roughly 50% clockwise across all trials (effectively to give rise to a flat psychometric function), they were removed from the analyses.

## 2.4 Questionnaire

All participants were asked to complete a comprehensive questionnaire. Importantly to the topic of this paper, participants were asked to self-identify their ethnicities as well their nationalities (the nation for which they claimed citizenship) with open ended questions. Two independent raters categorized participants' ethnicity as "East Asian", "white", or "other" based on their written responses. Those categorized as "white" included those who self-identified as "white", "Caucasian", and with other European ancestry. Those categorized as "East Asian" included those who self-identified as Asian, Chinese, Taiwanese, Japanese, Korean, and Vietnamese. All participants outside of these identifiers, including those who were multiracial, were categorized as "other" for this investigation. This questionnaire appeared at the very end of the experiment, so that these questions could not influence participants' performance on the RFT. If they identified themselves as a United States (U.S.) citizen, participants were asked to provide generational information. That is, how many generations of their family had lived in the U.S. First generation meant that the participant was a citizen, but was not born in the U.S. Second generation was defined as being born in the U.S., but at least one parent was born in another country. Third generation was defined as being born in the U.S., both parents were born in the U.S., but at least one grandparent was born in another country.

## 3. Results

### 3.1 Alignment task

Out of 342 participants who completed the alignment RFT and the questionnaire, 207 participants identified themselves either as East Asian (n = 121; of which 66.9% were female) or white (n = 86; of which 63.9% were female), with a total of 136 female (66%), and 71 male participants. The East Asian participants mostly comprised of Chinese (49%) and South Koreans (16%). The remaining 135 participants were categorized as "other". Within the "other" category, 38.5% identified as Hispanic/Latinx, 23.7% as Middle Eastern (including Persian, Armenian, Pakistani, and Israeli), 13.3% as Native American, 3.7% as African American, and 20.7% as multiracial. Within this "other" category, 86 (64%) were female.

**3.1.1 SV bias.**   We averaged the SV biases of the two tilted frame conditions by first flipping the sign for the -18˚ tilt to compare to the non-tilted frame condition.

Using a 2 (Sex) × 2 (Ethnicity) × 2 (Frame) ANOVA on the bias data, we found an expected main effect of frame, $F$ (1, 203) = 223.72, $p$ < .001, $\eta_p^2 = .52$, indicating that a tilted frame produced biases towards the tilt of the frame ($M$ = 3.65˚, $SD$ = 2.81˚) compared to the non-tilted

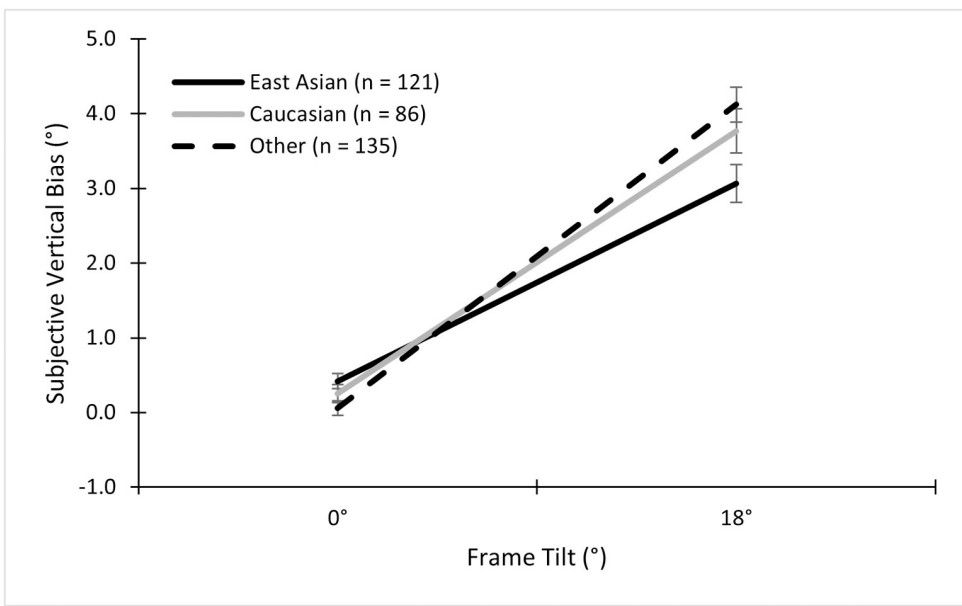

**Fig 3. Mean subjective vertical biases in the alignment task across ethnicities.** Mean subjective vertical biases for the non-tilted (0˚) and tilted (±18˚) conditions for East Asian, white, and other participants in the alignment RFT. Error bars represent standard errors, as they do in all subsequent graphs.

frame ($M$ = 0.25˚, $SD$ = 1.15˚). There was also a main effect of sex, $F$ (1, 203) = 9.99, $p$ = .002, $\eta_p^2 = .05$. This effect was driven by differences between women and men in the tilted frame conditions, as suggested by the interaction found between frame and sex, $F$ (1, 203) = 21.53, $p < .001, \eta_p^2 = .10$. Specifically, women tended to have a slightly larger bias towards the tilt of the frame ($M$ = 4.34˚, $SD$ = 2.69˚) than men ($M$ = 2.96˚, $SD$ = 2.68˚). There was also an interaction effect between frame and ethnicity, $F$ (1,203) = 4.45, $p$ = .036, $\eta_p^2 = .02$. This interaction suggested that white participants were more biased by a tilted frame (Fig 3). However, the effect size was small. Importantly, there was no main effect of ethnicity, $F$ (1, 203) = 1.91, $p$ = .17; nor any interaction between ethnicity and sex, $F$ (1, 203) = 0.41, $p$ = .52; nor any three-way interaction, $F$ (1, 203) = 0.58, $p$ = .45.

Since no ethnicity effect was found, we investigated the effects of sex on the frame bias by including all participants (222 females, 120 males) in a 2 (Frame) × 2 (Sex) and found similar results as reported above. There was a main effect of the frame tilt, $F$ (1, 340) = 428.92, $p <$ .001, $\eta_p^2 = .56$; a main but small effect of sex, $F$ (1, 340) = 11.06, $p$ = .001, $\eta_p^2 = .03$; and an interaction of sex and frame tilt, $F$ (1, 340) = 22.41, $p < .001, \eta_p^2 = .06$ (see Fig 4).

**3.1.2 Generational data.** Out of the 121 East Asian participants, 64 were 2<sup>nd</sup> generation Americans or beyond, while 57 were 1<sup>st</sup> generation Americans or were not U.S. citizens. Out of the 86 white participants, 73 were 2<sup>nd</sup> generation Americans and beyond, and 13 were 1<sup>st</sup> generation or non-citizens. We did not find any generational effects when testing a 2 (Generation) × 2 (Ethnicity) × 2 (Frame) ANOVA on biases. Further, within only the East Asian participants using a 2 (Generation) x 2 (Frame) ANOVA, we only found the expected effect of frame, $F$ (1, 119) = 137.94, $p < .001, \eta_p^2 = .54$. The main effect of generation and interaction effect were not significant, $F$ (1,119) = 0.38, $p$ = .54; $F$ (1,119) = 0.16, $p$ = .69, respectively.

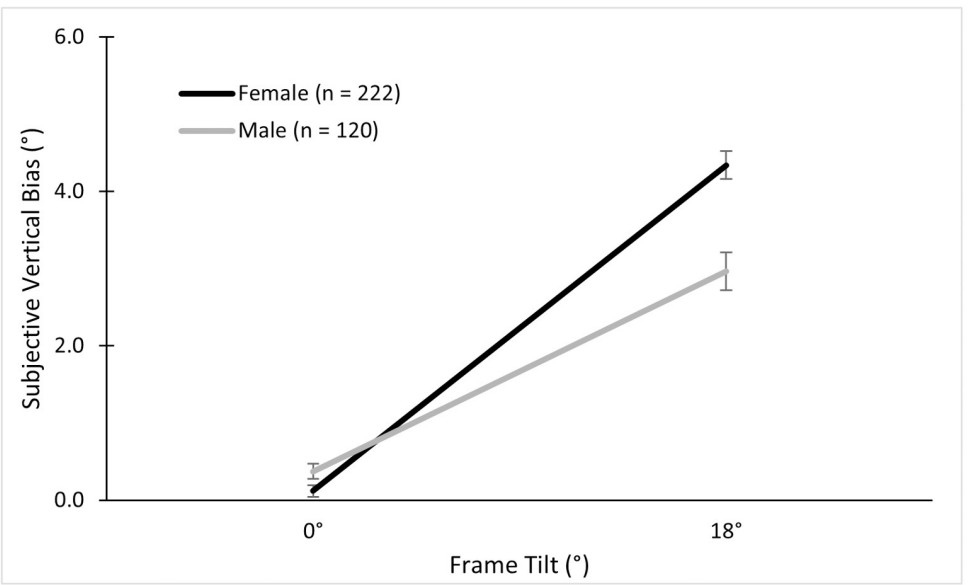

**Fig 4. Mean subjective vertical biases in the alignment RFT across sex.**

## 3.2 Discrimination task

Out of 220 people who completed the orientation discrimination RFT above chance and the questionnaire, 141 identified themselves either as East Asian (n = 84; of which 71.4% were female) or white (n = 57; of which 61.4% were female); with 95 females and 46 males. The remaining 79 participants were categorized as "other" (60.8% were female). Within the "other" category, 34.2% identified as Hispanic/Latinx, 27.8% as Middle Eastern, 7.6% as Native American, 7.6% as African American, and 22.8% as multiracial.

**3.2.1 Bias.** Using a 2 (Ethnicity) × 2 (Sex) × 2 (Frame) ANOVA, we found the expected main effect of frame, $F(1, 137) = 211.34$, $p < .001$, $\eta_p^2 = .61$. Bias was greater towards the tilt of the frame when the frame was tilted ($M = 4.96°$, $SD = 3.25°$) compared to when the frame was not tilted ($M = 0.60°$, $SD = 1.29°$). We also found the main effects of sex, $F(1, 137) = 14.76$, $p < .001$, $\eta_p^2 = .10$; and of ethnicity, $F(1, 137) = 4.76$, $p = .031$, $\eta_p^2 = .03$, such that females and white participants showed greater biases. There was a significant interaction between frame and sex suggesting a difference specifically in the tilted frame condition, with women reporting greater biases ($M = 6.16°$, $SD = 3.12°$) than men ($M = 3.75°$, $SD = 2.98°$), ($F(1, 137) = 18.68$, $p < .001$, $\eta_p^2 = .12$). Additionally, a significant interaction between frame and ethnicity suggested that white participants had greater biases in the tilted frame condition ($M = 5.69°$, $SD = 3.09°$) than East Asian participants ($M = 4.22°$, $SD = 3.32°$), ($F(1, 137) = 7.92$, $p = .006$, $\eta_p^2 = .06$), see Fig 5. There was no interaction between sex and ethnicity, $F(1, 137) = 0.02$, $p = .90$; nor was there a three-way interaction, $F(1, 137) = 0.06$, $p = .92$.

The effect of sex persisted in the larger sample (143 females, 77 males), as found in an ANOVA with data from all participants. There was a main effect of the frame tilt, $F(1, 218) = 345.63$, $p < .001$, $\eta_p^2 = .61$. Bias was greater towards the tilt of the frame when the frame was tilted ($M = 5.37°$, $SD = 3.50°$) compared to when the frame was not tilted ($M = 0.49°$, $SD = 1.26°$). We again found a main effect of sex, $F(1, 218) = 12.39$, $p = .001$, $\eta_p^2 = .05$; and an interaction of sex and frame tilt, $F(1, 218) = 15.44$, $p < .001$, $\eta_p^2 = .07$. The effect of sex

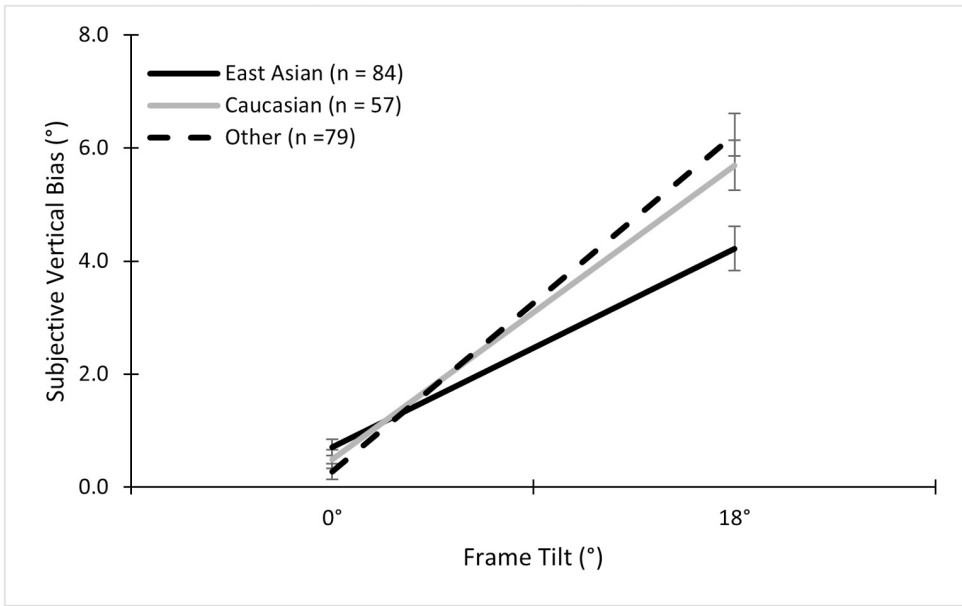

**Fig 5. Mean subjective vertical biases in the discrimination task across ethnicities.**

suggested a difference specifically in the tilted frame condition, with women reporting greater biases ($M = 6.26°$, $SD = 3.33°$) than men ($M = 4.49°$, $SD = 3.33°$), see Fig 6.

**3.2.2 Standard deviation data (1/slope of the psychometric function, or 1/sensitivity).** Using a 2 (Sex) × 2 (Ethnicity) × 2 (Frame) on standard deviations of the Gaussian that fitted each participant's psychometric function, we found a main effect of frame, $F(1, 137) = 215.69$, $p < .001$, $\eta_p^2 = .61$. This means that, when the frame was upright, participants were better able to tell whether the rod was left or right tilted from vertical, than when the frame was tilted

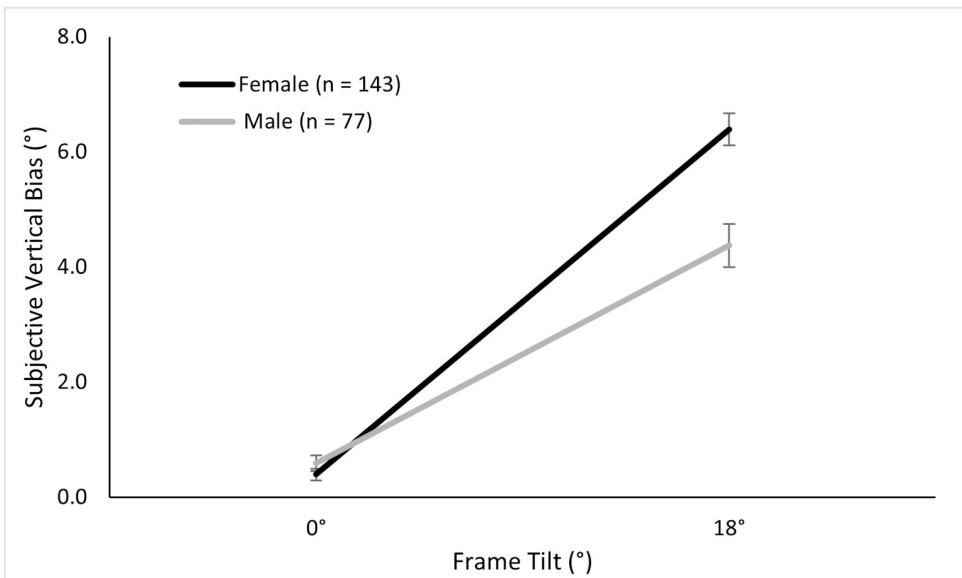

**Fig 6. Mean subjective vertical biases in the discrimination RFT across sex.**

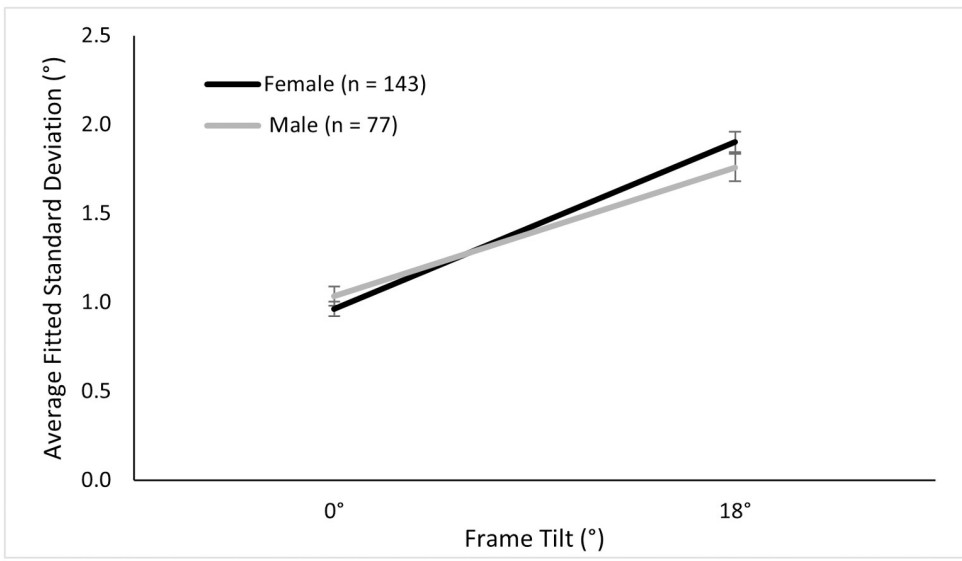

**Fig 7. Mean standard deviations across trials in the discrimination task.** Mean standard deviations of the fitted Gaussian for the tilted and non-tilted conditions for females and males in the discrimination RFT.

(Fig 7). The interaction between frame and sex was marginally significant, $F(1, 137) = 3.67$, $p = .058$, $\eta_p^2 = .03$, suggesting that women had slightly less discrimination sensitivity than men, when the frame was tilted. In order to further verify, we compared men and women in the larger sample in a 2 (Frame) × 2 (Sex) ANOVA. We found a similar trend. Namely, there was a main effect of the frame tilt, $F(1, 218) = 372.04$, $p < .001$, $\eta_p^2 = .63$; and an interaction of sex and frame tilt, $F(1, 218) = 6.17$, $p = .014$, $\eta_p^2 = .03$, see Fig 6. (The main effect of sex was not significant, $F(1, 218) = 0.21$, $p = .65$.)

**3.2.3 Generational data.** Out of the 84 East Asian participants, 40 were 2nd generation Americans or beyond, while 44 were 1st generation or non-citizens. Out of the 57 white participants, 51 were 2nd generation or beyond, and six were 1st generation or non-citizens. When testing a 2 (Generation) × 2 (Ethnicity) × 2 (Frame) ANOVA, we found main effects of frame as expected, $F(1, 137) = 96.27$, $p < .001$, $\eta_p^2 = .41$; and of generation, such that those later-generation Americans had greater biases than the 1st generation or non-citizens, $F(1, 137) = 5.70$, $p = .018$, $\eta_p^2 = .04$. We found a three-way interaction between frame, ethnicity, and generation, $F(1, 137) = 4.09$, $p = .045$, $\eta_p^2 = .03$. The three-way interaction suggests that within the non-tilted frame condition, white and East Asian participants both had similar biases regardless of generation. In contrast, in the tilted frame conditions, East Asians tended to have similar biases regardless of generation, while white participants had greater biases if they were later generations. However due to the small sample of six white participants who fell into the non-citizen/1st generation category, this interaction effect that contain both ethnicity and generation should be viewed cautiously. We also found a two-way interaction between generation and frame, suggesting that, when the frame was tilted, the later generations had greater biases, $F(1, 137) = 4.82$, $p = .03$, $\eta_p^2 = .03$, see Fig 8. We further found a significant interaction between generation and ethnicity, suggesting that within white participants, the later generations had greater biases. In comparison, the East Asians showed similar biases across generations, $F(1, 137) = 3.50$, $p = .063$, $\eta_p^2 = .03$ (see Fig 9).

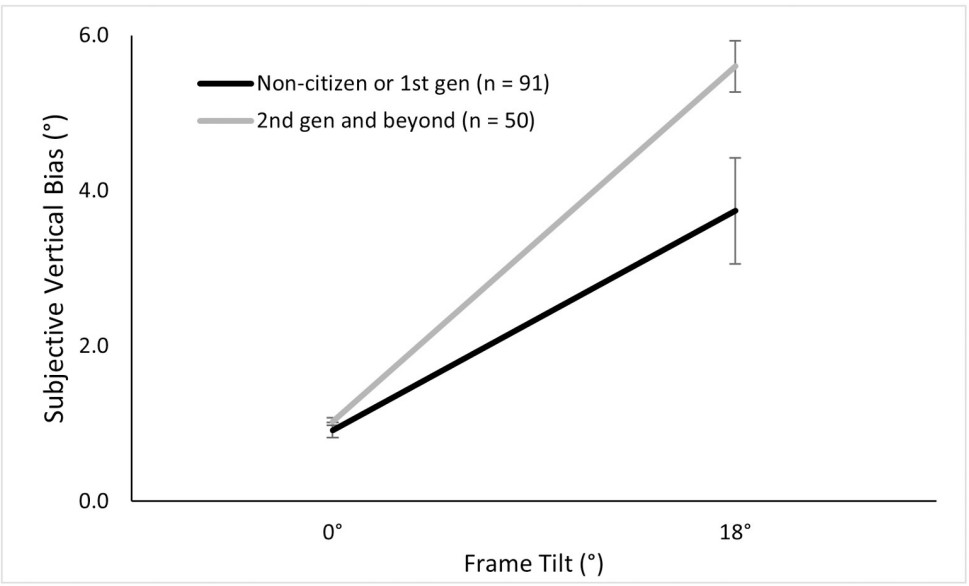

**Fig 8. Mean subjective vertical bias across generations and tilt conditions.**

The interaction effect was unexpected according to the culture theory under discussion, because the later generation Asian Americans were expected to reduce their frame-induced bias that should be closer to the white Americans', while the white Americans were expected to maintain their low bias throughout. Due to this unexpected result, a within-Asian ANOVA was investigated further with 2 (Generation) × 2 (Frame) factors. We found the expected main effect of frame, $F(1, 137) = 122.52$, $p < .001$, $\eta_p^2 = .60$. Consistent with the earlier three-way ANOVA above, we found no main effect of generation, $F(1, 82) = 0.34$, $p = .56$; nor any interaction between frame and generation, $F(1, 82) = 0.04$, $p = .84$, see Fig 10.

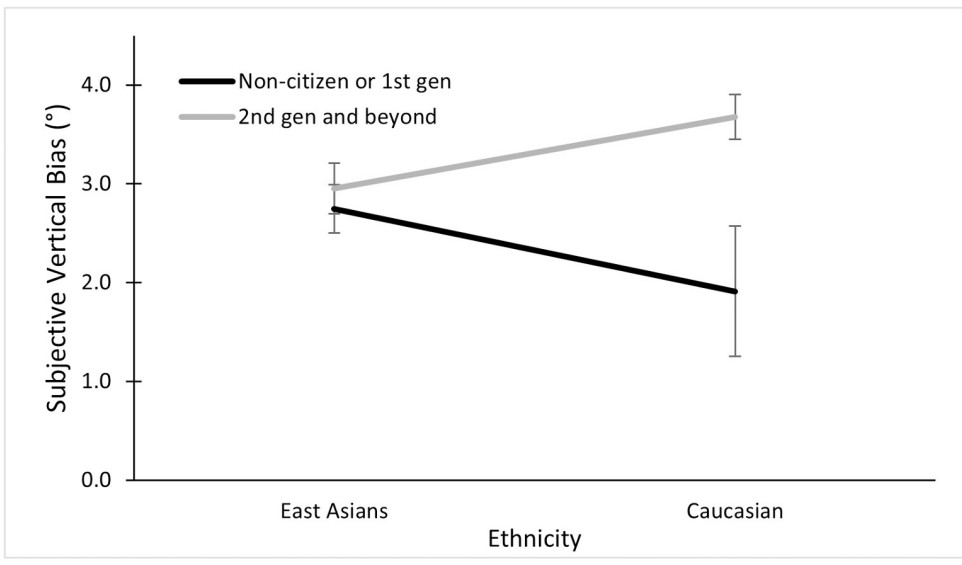

**Fig 9. Overall mean subjective vertical bias across generations, collapsed across tilt conditions.**

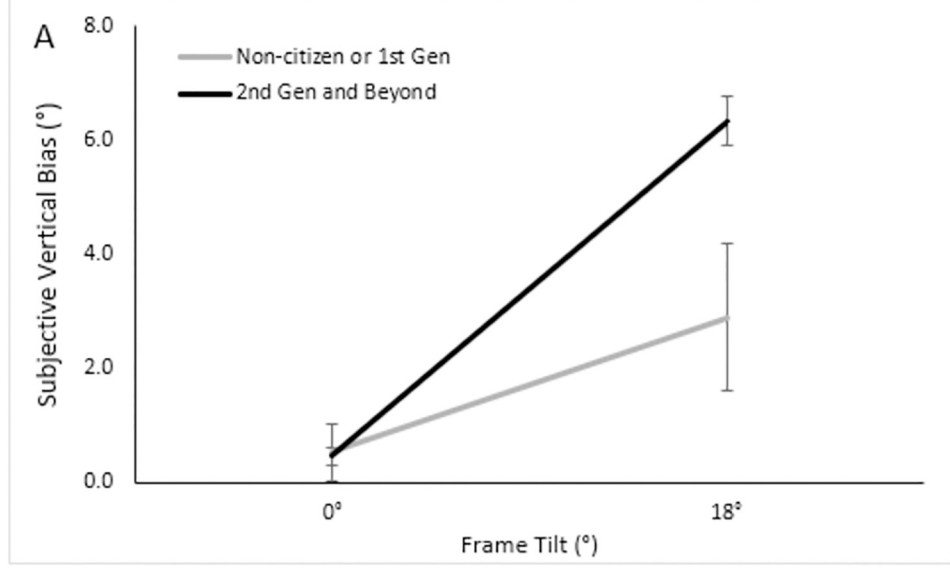

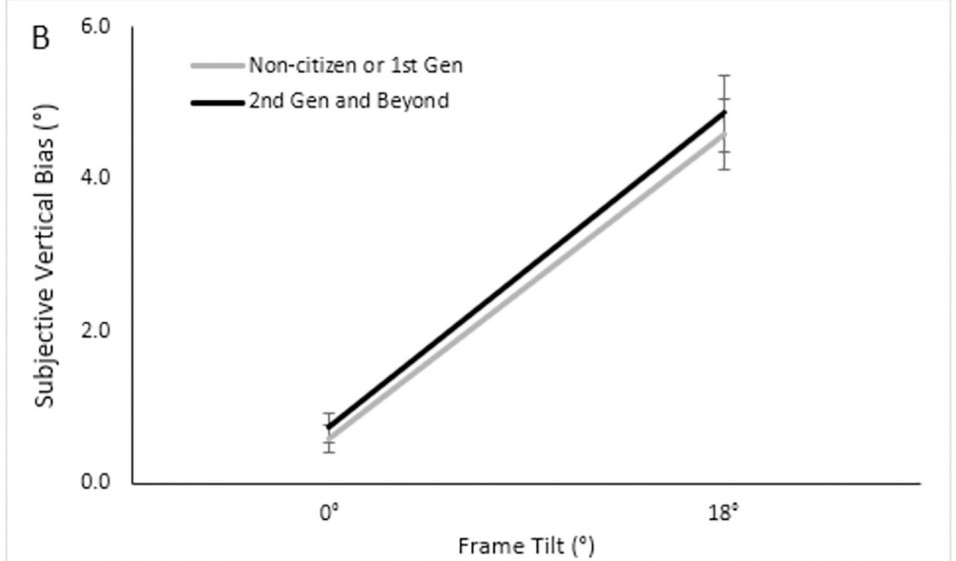

**Fig 10. Mean subjective vertical biases across tilt conditions between ethnicities.** Mean subjective vertical biases across tilt conditions for white participants (Panel A) and East Asian participants (Panel B), as compared between non-citizens and 1st generation to 2nd generation and beyond.

We further looked into the fitted standard deviation data and found a three-way interaction using a 2 (Generation) × 2 (Ethnicity) × 2 (Frame) ANOVA, $F$ (1, 137) = 5.18, $p$ = .024, $\eta_p^2$ = .04. Namely, no difference was apparent when the frame was upright (see Fig 11, panel A). When the frame was tilted, however, the 1st generation / non-citizen Asians had a lower discrimination sensitivity than their white counterparts (see Fig 11, panel B). This is to say, the 1st generation / non-citizen Asians had more difficulty deciding whether the rod was tilted left or right with respect to their perceived vertical, when the frame itself was tilted.

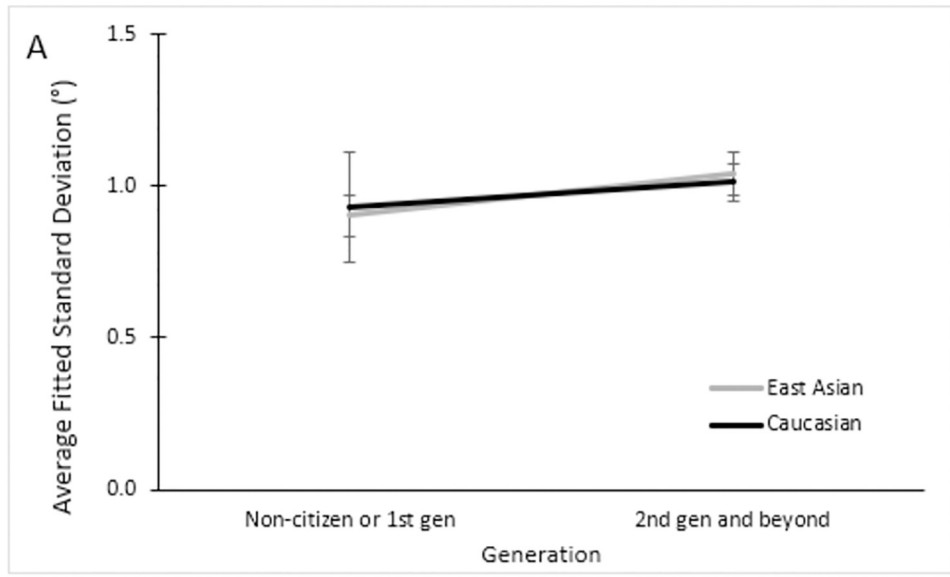

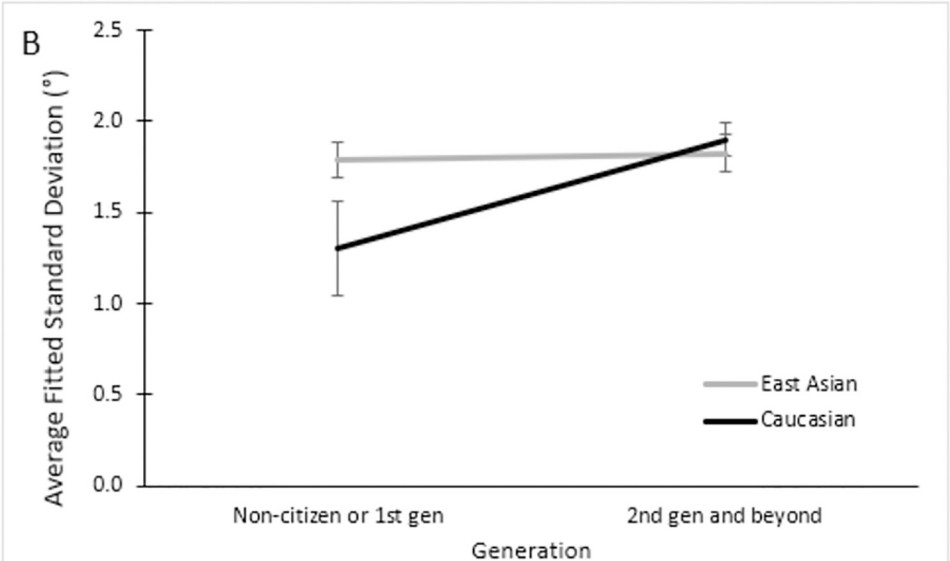

**Fig 11. Mean standard deviations across tilt conditions between ethnicities.** Mean standard deviations of the fitted Gaussian across ethnicities in the non-tilted condition (Panel A) and the tilted condition (Panel B) comparing non-citizens and 1st generation to 2nd generation and beyond.

## 4. Discussion

Is a person's perception and cognition influenced by the person's cultural background? Many publications in the literature, as far as we know, are affirmative to this question. This question also remains current and active, as indicated by the following recent findings. For example, [17] used the classic visual search paradigm to first confirm that searching for a longer line among shorter lines was faster than vice versa for North Americans. In comparison, such search asymmetry was absent for Japanese participants. [18] compared between Canadian and Chinese participants in face recognition and found that Canadians relied more on higher spatial frequencies of the faces whereas the Chinese on lower spatial frequencies. This result

implied that the Chinese participants spatially attended more broadly than their Canadian counterparts during visual face processing. By extension, this result is also consistent with the results in [6] that Easterners attended more to spatial context and therefore were influenced more by the same spatial context during processing of the rod's orientation. While the current study was not intended to be a replication of [6], to our knowledge their study is the closest in focus to the current investigation. Thus we made some direct comparisons to understand the differences in results. We did not find their previously reported differences between East Asian participants and white participants.

The closest piece of evidence we found that was consistent with the culture account, was the finding that Asian participants had lower discrimination sensitivity since it implied that Asian participants were more influenced by the spatial context of the focal object than their white counterparts. However, as far as we know, the culture account only predicted the bias but made no prediction regarding discrimination sensitivity. Moreover, according to the culture account, white participants were not expected to change their discrimination abilities across generations (it got worse in the data), while the Asian participants were expected to improve their discrimination across generations in the U.S. (it did not change in the data). Taken together, we found no strong evidence that a participant's cultural background had anything to do with judging a rod's orientation inside a tilted frame. However, it does not appear to be the case that our data were simply noisier than those in other studies, for the following reasons.

1. We were able to replicate similar biases and variances of the RFT in aspects unrelated to participant cultures. Note, however, the virtual RFT used in this experiment replicated other well-documented effects in the literature. The average bias on the RFT in the non-tilted condition was less than 1˚, typical of conditions without a frame or with a non-tilted frame in healthy participants [19–21]. The average error found for the tilted frame conditions, 3.65˚, was similar in magnitude to what others have found in 3D virtual and real RFTs, ranging from 3˚ to 5˚ in young adults (e.g., [10, 13, 22]). Further, we replicated the well-known finding in the literature that women were slightly (but with statistical significance) more influenced than men by the tilted frame in judging the rod's orientation. This indicated that our data were not noisier than usual and not merely giving rise to effects that might be due to random variations of data.

2. Our data showed a small yet statistically significant interaction effect between culture and generation. Following the cultural account, it was predicted that Westerners in the U.S. should show little generational shift, whereas Easterners in the U.S. should shift toward their Western counterparts in visual processing. One limitation of the current study is that the length of stay in the U.S. was not collected from those who were non-citizens. As mentioned in the introduction, the effect of culture may be lessened between white and East Asian participants based on the length of stay and immersion of East Asian participants in the westernized culture of the U.S. In the current study, given the overall large sample size, even if the effect was lessened, statistical power was sufficient to detect even small differences between the cultures. Additionally, the small effect that we did observe was opposite of what has been discovered previously, suggesting more research should be conducted in this area. However, we do not have any theoretical hypothesis as to why our data showed such a pattern. Given the small sample size in some generation conditions, this effect should be taken with caution.

3. Another aspect not manipulated in the current study was the amount of control our participants had over their responses. [6] tested whether the amount of control the participants

had over the adjustment of the rod influenced participants' performance. They found that American males benefited from controlling the rotation of the rod in the RFT, while East Asian males' performance decreased. In our experiment, all participants were in control of the rotation in the alignment RFT. This should have produced the best possible performance of the white males. However, our data showed the opposite, that East Asian males performed slightly better than white males. Also, in our study, white participants got worse in their discrimination abilities across generations, while their Asian counterparts did not change. We do not have an explanation why our data contradicted theirs or, more generally, why we could not replicate their results. Apparently, our virtual reality experiment was different from theirs using a physical RFT apparatus. Our participants also stood on a foam mattress while performing the RFT. It is unclear whether these methodologic differences could be responsible for the different results. However, as noted above, we have replicated the frame effects similar in magnitude to those found in other studies using the physical apparatus and a 3D virtual RFT. Regardless, these methodological differences are worth looking into. But it seems fair to say that the cultural effects found earlier may not be as robust as previously thought. We found that females showed a larger bias than males in judging the rod's vertical orientation, when the frame was tilted. This finding was consistent with previous findings in the literature [12]. However, given that sex difference was not a main focus of the current study, we did not collect any menstrual phase information from our female participants to correlate between the size of the bias and menstrual phase (as a proxy of sex hormones), as did in [12]. As such, this finding in the current study served only as confirmation that the non-cultural aspects of our data were consistent with the literature, giving some credence to the quality of our data.

We believe that the best way forward to resolve the above discrepancies is additional data from well-controlled experiments. Given the scope of potential cultural effects on perception and cognition, these are important areas of investigation and consideration for researchers in the field.

## Author Contributions

**Conceptualization:** Zili Liu.

**Data curation:** Chéla R. Willey.

**Formal analysis:** Chéla R. Willey.

**Investigation:** Chéla R. Willey.

**Methodology:** Chéla R. Willey, Zili Liu.

**Project administration:** Chéla R. Willey.

**Supervision:** Zili Liu.

**Writing – original draft:** Chéla R. Willey, Zili Liu.

**Writing – review & editing:** Zili Liu.

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
