## [Decision Letter · Decision Letter 0]

4 Jul 2022

PONE-D-22-10991Re-assessing the role of culture on the visual orientation perception of the rod and frame testPLOS ONE

Dear Dr. Willey,

Thank you for submitting your manuscript to PLOS ONE. After careful consideration, we feel that it has merit but does not fully meet PLOS ONE’s publication criteria as it currently stands. Therefore, we invite you to submit a revised version of the manuscript that addresses the points raised during the review process.

We look forward to receiving your revised manuscript.

Kind regards,

Alastair Smith

Academic Editor

PLOS ONE

Journal Requirements:

If you are reporting a retrospective study of medical records or archived samples, please ensure that you have discussed whether all data were fully anonymized before you accessed them and/or whether the IRB or ethics committee waived the requirement for informed consent. If patients provided informed written consent to have data from their medical records used in research, please include this informatio

3. Please note that according to our submission guidelines (http://journals.plos.org/plosone/s/submission-guidelines), outmoded terms and potentially stigmatizing labels should be changed to more current, acceptable terminology. To this effect,  “Caucasian” should be changed to “white” or “of [Western] European descent” (as appropriate).

Reviewers' comments:

Reviewer's Responses to Questions

**Comments to the Author**

1. Is the manuscript technically sound, and do the data support the conclusions?

Reviewer #1: Partly

Reviewer #2: Partly

2. Has the statistical analysis been performed appropriately and rigorously? 

Reviewer #1: Yes

Reviewer #2: Yes

3. Have the authors made all data underlying the findings in their manuscript fully available?

Reviewer #1: Yes

Reviewer #2: Yes

4. Is the manuscript presented in an intelligible fashion and written in standard English?

Reviewer #1: Yes

Reviewer #2: Yes

5. Review Comments to the Author

Reviewer #1: This study addressed cross-cultural differences in perception, specifically in visual perception, by using a visuospatial task that assesses the person's ability to align a rod to true vertical (gravity) or identify the direction of the rod when surrounded by a distracting tilted frame. The manuscript is generally well written, but there are a few concerns.

1. The authors can still enrich their Introduction with information on how culture influences attentional processes (direct attention to focal objects or contexts), and on the type of processing (analytical or holistic) different cultures prefer.

2. In the Results section, when referring to "Others", it would be useful to include what is meant by that. It is not certain whether the authors included this information in the included appendix of the questionnaire.

3. Also for "Others", it would make it easier for the reader if the gender distribution is specified for this group as well at each analysis trial.

4. For generational data, it is not clear how long the 1st generation Americans or the non-US citizens have been in the USA. It is possible that the time of their stay may influence the extent to which they were affected by the culture they are exposed to.

5. Line 223: it is not clear what is meant by " and interestingly since it was in the "wrong" direction of ethnicity".

6. Lines 239, 240. This is a run-on sentence.

7. The figures are not numbered. They should be.

8. In the figure on generation effects (I assume it is Figure 8), the authors should include number of subjects (n).

9. It would be worth adding how sex hormones influence performance on visuospatial tasks either in the Introduction or Discussion to delineate any gender effects.

Reviewer #2: This is a short and interesting study but some elements, which are overall minor in my opinion, should be improved before it can get published. Below are my point-by-point comments about the paper.

Line 83. Is there a justification for using a 3D version of the RFT here? Is it because it may increase the effect size and therefore be more sensitive to detect cultural variations?

Line 83. How is culture operationalised and tested? Is culture tested in different countries? (one discovers that this is not the case when reading the paper, but this should be prepared). Is there a justification/good reason for testing culture only within the US? One may refer to studies which found that cultural effects are smaller/decrease for Asians living in the US, or Amercians living in Asia (e.g., Kitayama,Duffy,Kawamura, and Larsen, 2003)

Line 84. The meaning of "to anticipate" here is difficult to grasp. I had to read the sentence twice to understand its meaning. It was not clear to me, at first, that this was a preview of all the findings of the study.

Line 111. Lack of clarity, the sentence suggests that there were three conditions (-18°, 0°, 18°), with 6 trials each. Is this correct? If yes, could this be said explicitly?

Line 158. A description of how culture is defined, in subsequent analyses, would be useful. In addition, given that culture is the main IV in this study, it is important to provide specific information about the questions asked to the participants in order to classify them.

Line 166. Who are the 137 participants who do not identify as EA or Caucasian? How is Caucasian actually defined? Does it mean white skin? What about participants of mixed origins, are they other? We are given little information about the main IV of the study, and this is problematic.

Line 179. Does Ethnicity only have two levels? What about the 'other' group shown on the graphs? What does it mean exactly?

Line 190. Although the main effect of ethnicity is not significant, there may be a trend there. Same comment as below.

Line 195. The main effect of Frame suggests that overall performance varied across males and females. Were RTs measured, in ordre to check for possible engagement effects? (i.e., longer RTs, coupled with better overall performance and lower illusion size?)

Line206. How is a second generaion Asian defined? Is it enough to have one (out of 4) grandparents of Asian decent to be considered a 2nd-generation Asian?

Line 314. This conclusion is somewhat surprising. A number of analyses suggest that the performance of the two cultural groups differ. In fact, there is no strong evidence for an absence of difference. the fact that the difference does not go in the direction predicted by what is called the 'culture theory' (making it sound as though there is only one possible account for cultural effects), does not necessarily mean that there is no effect. Rather, it may mean that if the effect shown here actually exists (and one cannot really decide one way or another), it cannot be accounted for by the social organisation theory.

Line 373. The discussion feels short, with no attempt to account for the cultural effect observed in the study, and little discussion about the gender effect observed.

6. PLOS authors have the option to publish the peer review history of their article (what does this mean?). If published, this will include your full peer review and any attached files.

Reviewer #1: No

Reviewer #2: No

---

## [Author Response · Author response to Decision Letter 0]

23 Aug 2022

The responses to reviews are uploaded in a separate file.

---

## [Decision Letter · Decision Letter 1]

28 Sep 2022

PONE-D-22-10991R1Re-assessing the role of culture on the visual orientation perception of the rod and frame testPLOS ONE

Dear Dr. Liu,

Thank you for submitting your revised manuscript to PLOS ONE. Both reviewers are satisfied that their comments have been addressed, although you will see that Reviewer #1 has a couple of remaining points that are worth considering. I would like to invite you to address these in a minor revision. If I am satisfied that they have been appropriately handled then we may not require another round of reviews (although, of course, this cannot be guaranteed).

We look forward to receiving your revised manuscript.

Kind regards,

Alastair D. Smith

Academic Editor

PLOS ONE

Journal Requirements:

Reviewers' comments:

Reviewer's Responses to Questions

**Comments to the Author**

1. If the authors have adequately addressed your comments raised in a previous round of review and you feel that this manuscript is now acceptable for publication, you may indicate that here to bypass the “Comments to the Author” section, enter your conflict of interest statement in the “Confidential to Editor” section, and submit your "Accept" recommendation.

Reviewer #1: All comments have been addressed

Reviewer #2: All comments have been addressed

2. Is the manuscript technically sound, and do the data support the conclusions?

Reviewer #1: Partly

Reviewer #2: Yes

3. Has the statistical analysis been performed appropriately and rigorously? 

Reviewer #1: Yes

Reviewer #2: Yes

4. Have the authors made all data underlying the findings in their manuscript fully available?

Reviewer #1: Yes

Reviewer #2: Yes

5. Is the manuscript presented in an intelligible fashion and written in standard English?

Reviewer #1: Yes

Reviewer #2: Yes

6. Review Comments to the Author

Reviewer #1: The authors have addressed my previous comments, but I have a few more for this revised manuscript.

Overall, I feel that cultural studies should include participants within their cultural habitat for comparisons sake, rather than including participants of different generations in a country different than their own. Having said that, this is no implication to the significance of this study.

1. In the updated abstract, please use "white Western Europeans", rather than "white" participants, because at this point of the manuscript, the term has not been defined yet.

2. Page 5, lines 89-90: Please put the reference [9] after the 2D computerized test, not in line 90 citation [9, 10].

3. I feel that the authors included too many comments and even reference citations in the Results sections (3.1.1 SV Bias; 3.2.3 Generational Data); it is better to just report the findings in the Results section, and then leave the explanations for the Discussion section.

4. Why is the number of participants for Generational Data different in 3.1.2 and 3.2.3?

5. The authors state that they cannot offer an explanation for their paradoxical findings which contradicted their hypothesis and those from a previous study [Reference 6] that reported that East Asians were more prone to larger errors on the RFT, in comparison to their Western counterparts. Unexpectedly, other results also indicate that later generation Americans, which are quite immersed in American culture, displayed greater biases than the 1st generation Americans or non-citizens.

Whether methodology differences might have affected the results, specifically the RFT protocol, is not clear. Unlike Reference 6, participants in this study stood on a foam block while aligning the rod to vertical. This might be worth looking into.

Reviewer #2: I find that the revised version of the manuscript adequately addresses all the issues raised during the review.

7. PLOS authors have the option to publish the peer review history of their article (what does this mean?). If published, this will include your full peer review and any attached files.

Reviewer #1: No

Reviewer #2: No

---

## [Author Response · Author response to Decision Letter 1]

30 Sep 2022

Reviewer #1: The authors have addressed my previous comments, but I have a few more for this revised manuscript.

Overall, I feel that cultural studies should include participants within their cultural habitat for comparisons sake, rather than including participants of different generations in a country different than their own. Having said that, this is no implication to the significance of this study.

1. In the updated abstract, please use "white Western Europeans", rather than "white" participants, because at this point of the manuscript, the term has not been defined yet.

Thanks, abstract was edited per the suggestion.

2. Page 5, lines 89-90: Please put the reference [9] after the 2D computerized test, not in line 90 citation [9, 10].

Fixed, reference moved per the suggestion. 

3. I feel that the authors included too many comments and even reference citations in the Results sections (3.1.1 SV Bias; 3.2.3 Generational Data); it is better to just report the findings in the Results section, and then leave the explanations for the Discussion section.

We edited these sections to move some of the commentary from the Results section into the Discussion section, which helped to reduce redundancy.

4. Why is the number of participants for Generational Data different in 3.1.2 and 3.2.3?

Section 3.1 describes data from all the participants who completed the alignment rod and frame task. Section 3.2 describes the data from the subset of participants who also completed the rod and frame discrimination task. 

5. The authors state that they cannot offer an explanation for their paradoxical findings which contradicted their hypothesis and those from a previous study [Reference 6] that reported that East Asians were more prone to larger errors on the RFT, in comparison to their Western counterparts. Unexpectedly, other results also indicate that later generation Americans, which are quite immersed in American culture, displayed greater biases than the 1st generation Americans or non-citizens.

Whether methodology differences might have affected the results, specifically the RFT protocol, is not clear. Unlike Reference 6, participants in this study stood on a foam block while aligning the rod to vertical. This might be worth looking into.

Thank you. We have revised our discussion per your suggestion, as follows (lines 425 – 437):

In our experiment, all participants were in control of the rotation in the alignment RFT. This should have produced the best possible performance of the white males. However, our data showed the opposite, that East Asian males performed slightly better than white males. Also, in our study, white participants got worse in their discrimination abilities across generations, while their Asian counterparts did not change. We do not have an explanation why our data contradicted theirs or, more generally, why we could not replicate their results. Apparently, our virtual reality experiment was different from theirs using a physical RFT apparatus. Our participants also stood on a foam mattress while performing the RFT. It is unclear whether this methodologic difference could be responsible for the different results. However, as noted above, we have replicated the frame effects similar in magnitude to those found in other studies using the physical apparatus and a 3D virtual RFT. Regardless, these methodological issues are worth looking into. But it seems fair to say that the earlier cultural effects may not be as robust as previously thought.

Reviewer #2: I find that the revised version of the manuscript adequately addresses all the issues raised during the review.

---

## [Editor Report · Decision Letter 2]

6 Oct 2022

Re-assessing the role of culture on the visual orientation perception of the rod and frame test

PONE-D-22-10991R2

Dear Dr. Liu,

Thank you for submitting your revised manuscript. I am satisfied that the amendments you made adequately address the final comments and am, therefore, pleased to formally accept your submission for publication once it meets all outstanding technical requirements.

With kind regards,

Alastair D. Smith

Academic Editor

PLOS ONE
---

## [Editor Report · Acceptance letter]

13 Oct 2022

PONE-D-22-10991R2 

Re-assessing the Role of Culture on the Visual Orientation Perception of the Rod and Frame Test 

Dear Dr. Liu:

I'm pleased to inform you that your manuscript has been deemed suitable for publication in PLOS ONE. Congratulations! Your manuscript is now with our production department. 

Kind regards, 

on behalf of

Dr Alastair Smith 

Academic Editor

PLOS ONE